# Mechanistic understanding of speciated oxide growth in high entropy alloys

Bharat Gwalani[1] ✉, Andrew Martin[1], Elizabeth Kautz[2,3], Boyu Guo [1], S. V. Lambeets [4], Matthew Olszta[2], Anil Krishna Battu[5], Aniruddha Malakar[1], Feipeng Yang [6], Jinghua Guo [6], Suntharampillai Thevuthasan[4], Ruipeng Li[7], Aram Amassian [1], Martin Thuo [1] & Arun Devaraj [4] ✉

Complex multi-element alloys are gaining prominence for structural applications, supplementing steels, and superalloys. Understanding the impact of each element on alloy surfaces due to oxidation is vital in maintaining material integrity. This study investigates oxidation mechanisms in these alloys using a model five-element equiatomic CoCrFeNiMn alloy, in a controlled oxygen environment. The oxidation-induced surface changes correlate with each element's interactive tendencies with the environment, guided by thermodynamics. Initial oxidation stages follow atomic size and redox potential, with the latter becoming dominant over time, causing composition inversion. The study employs in-situ atom probe tomography, transmission electron microscopy, and X-ray absorption near-edge structure techniques to elucidate the oxidation process and surface oxide structure evolution. Our findings deconvolute the mechanism for compositional and structural changes in the oxide film and will pave the way for a predictive design of complex alloys with improved resistance to oxidation under extreme conditions.

The concept of mixing equal or near equal amounts of elements to fabricate new alloys has sparked worldwide interest in alloy design. The approach is based on mixing multiple principal elements (MPE) in relatively high (often equiatomic) concentrations and stands in sharp contrast to the traditional practice of adding minor alloying elements. Though much emphasis is given to enhancing the mechanical properties while designing MPE alloys (MPEAs), understanding the corrosion and oxidation behavior of these alloys is an emerging research area[1-5]. The MPEAs based on transition elements, such as CoCrFeNiMnAl (referred to as high entropy alloys or HEAs), have shown improved combinations of strength, ductility, and strain hardenability[6,7]. The structure and stability of HEAs surface oxides, however, are not well understood largely due to a wide variety of compositions and limited understanding of the underlying atomic-

scale mechanisms of the oxidation process. Oxide formation tendencies in a liquid solution can thermodynamically be roughly deduced by Ellingham diagrams. However, considering many of these alloys are supersaturated-solid solutions, the concomitant oxidation and phase transformation at elevated temperatures are difficult to unveil[2,8,9]. Understanding of such compositionally complex systems demands a multiscale analysis of the oxidative degradation mechanism underpinning their structural instability.

At temperatures exceeding 500 °C, the oxidation behavior of traditional and complex alloys (e.g., Ni-Cr, Ni-based superalloys, Fe-Cr, CoCrFeNiMn 'Cantor' alloy) has been extensively studied in oxygen-rich environments and over extended periods (from days to hundreds of days[5,10-20]). Key contributors to the process include species transport to the oxide/metal junction within the base alloy, movement

[1]North Carolina State University, Department of Materials Science and Engineering, Raleigh, NC 27695, USA. [2]Energy and Environment Directorate, Pacific Northwest National Laboratory, Richland, WA 99352, USA. [3]North Carolina State University, Department of Nuclear Engineering, Raleigh, NC 27695, USA. [4]Physical & Computational Sciences Directorate, Pacific Northwest National Laboratory, Richland, WA 99352, USA. [5]Earth and Biological Sciences Directorate, Pacific Northwest National Laboratory, Richland, WA 99352, USA. [6]Advanced Light Source, Lawrence Berkeley National Laboratory, Berkeley, CA, USA. [7]National Synchrotron Light Source II, Brookhaven National Laboratories, Upton, NY 11973, USA. ✉e-mail: bgwalan@ncsu.edu; arun.devaraj@pnnl.gov

across the oxide/metal interface, and passage through the oxide layer. For instance, atmospheric oxidation of CoCrFeNiMn-containing alloys results in a layered oxide film, with Cr-rich inner and Mn-rich outer oxides attributed to outward cation diffusion[14]. In a study by Kim et al.[5], the high-temperature oxidation of FeCrMnNiCo (900–1100 °C in 20% O2/80% N2 for 24 h) showed the formation of layered $Cr_2O_3$ and $Mn_2O_3$ oxides at 900 °C. This process led to Ni, Fe, and Co enrichment in the base alloy. At 1000–1100 °C, the outer $Mn_2O_3$ layer changed to $Mn_3O_4$, forming a $(Mn, Cr)_3O_4$ ternary oxide with Ni, Fe, and Co in a solid solution. The presence of Mn, known for its high mobility and low-energy oxide formation, can deplete Mn in the alloy, causing pore formation and accelerating oxide growth[21].

Traditionally, chromium content in alloys is crucial for improving oxidation and corrosion resistance, due to the formation of a protective chromia layer. High-temperature materials use passivating elements like Al, Cr, or Si to form protective oxides. Cr's outward diffusion is faster than oxygen's inward diffusion, so chromia scales usually grow outward, consisting of fractions of iron, nickel, and manganese. However, at temperatures above 900 °C, chromia can form volatile $CrO_3$. Aluminum, another effective alloying element, forms a protective $Al_2O_3$ layer, offering significant resistance to high-temperature oxidation and corrosion. $Al_2O_3$ scales grow slower than chromium oxide scales and are more stable, especially against aggressive carbon or sulfur species. However, the formation of alumina and consequent depletion of Al in the alloy could result in structural instability in the alloy[22]. The formation of oxide scales in HEAs containing multiple oxide-forming species is influenced by their intricate chemistries and microstructures. High-temperature, long-duration oxidation studies on bulk alloys typically reveal dominant oxide films composed of complex ternary and quaternary oxides. The polycrystalline nature of these alloys, combined with surface heterogeneity and phase instabilities at high temperatures, leads to highly variable oxide structures. These structures are the result of complex interactions. Bhargavi et al., in their review, summarize the observations made in different HEAs during high-temperature oxidation, and not surprisingly it is difficult to develop a theoretical understanding of trends. Without independently examining each of these variables, a comprehensive understanding of oxidation pathways and scale stabilities, and the subsequent ability to engineer their physical properties, could remain elusive[23].

A deeper understanding of miscibility rules (beyond Hume-Rothery)[24] is critically needed to predict the behavior of each principal element as they navigate through the associated complex thermodynamic free energy space under evolving stressors (here oxidants). Coupling (i) stoichiometry dictated chaos on exposure to an oxidant (statistical thermodynamics equal a priori probability) and (ii) preferential oxidation based on each element's cohesive energy, diffusivity, and redox potential (kinetic resolution), enables prediction of the oxidation behavior in an otherwise highly complex system such as HEA. Once an initial surface oxide layer is established, a process that stems from Fickian diffusion evolves from an Arrhenius (Cabrera-Mott)[19] into linear (XRB)[25] and eventually a complicated logarithmic (Wagner)[18,26] rate law regime (Eqs. 1–3, respectively) hence hierarchical organization and speciation occurs.

$$D_A = e^{\left(-\frac{\Delta E_a}{kT}\right)} \tag{1}$$

$$D_A = \frac{\Gamma a^2}{2} = \frac{a^2 \nu}{2} e^{\left(-\frac{\Delta E_a}{kT}\right)} \tag{2}$$

$$D_A = \frac{\mu kT}{Ze} \tag{3}$$

Where $D_A$ = diffusion coefficient, $\Delta E_a$ = energy barrier, $\Gamma$ = interfacial excess, a = ion jump distance, $\nu$ = frequency, $\mu$ = mobility, Z = valence of diffusing particle, and e = electron charge.

Under oxidizing environments and elevated temperatures, surface speciation is highly dictated by each element's reactivity towards the diffusing oxygen atoms (oxidation potential), albeit perturbed by their interaction with nearest neighbor atoms. Properties such as standard reduction potential ($E^0$), cohesive energy density (CED)[27,28], and atomic radius aid in delineating the propensity and feasibility of the oxidation reaction. Thermodynamically, elements with the closest and deepest local minima act spontaneously given favorable conditions.

In this work, we apply the preferential interactivity parameter (PIP) to map and predict the speciation and distribution of different elements within a HEA under an oxidizing environment (see Fig. 1a and Supplementary Notes 1, 2)[29,30]. A mechanistic understanding of these parameters creates a pathway toward designing robust alloys with superior bulk and surface properties. Supported by in situ atom probe tomography (APT), where a sharpened needle specimen is oxidized

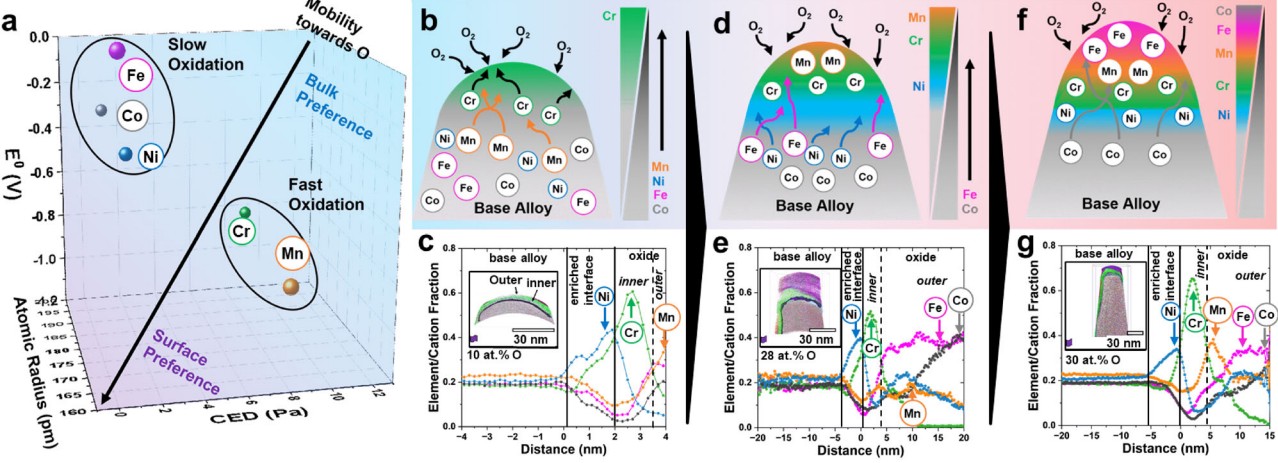

**Fig. 1 | Progression in speciation during oxidation of five component HEA.** **a** Preferential interactivity parameter (PIP) for the elements involved in the HEA for early prediction of oxidation behavior. **b** Schematic of the elemental distribution during initial oxidation at $t = 0$. **c** Atom probe proximity histograms (proxigrams) of HEA needle's top after oxidation at 400 °C for $t = 2$ min. **d** Schematic of the elemental distribution during initial oxidation at $0 < t < \infty$. **e** Atom probe proxigrams of HEA after oxidation at 400 °C for $t = 30$ min. **f** Schematic of the elemental distribution during initial oxidation at $t \sim \infty$. **g** Atom probe proxigrams of HEA after oxidation at 400 °C for $t = 120$ min. Further details on the APT experimental results can be seen in the following section.

within the APT instrument as a function of time, followed by ex situ high-resolution electron microscopy and X-ray diffraction and spectroscopy, a clearer image beyond the traditional two-dimensional view of speciated oxide growth is revealed. Our findings deconvolute the mechanism for compositional and structural changes in the oxide film assisting towards a predictive design of complex alloys with improved resistance to oxidation under extreme conditions.

## Results and discussion

Hypothetically, each element within an equiatomic HEA, such as CoCrFeNiMn, at time, $t = 0$, has equal thermodynamic probability to oxidize but a diffusion and redox-driven gradation immediately ensues at $t > 0$. Cr and Mn, exhibit the lowest $E^0_{Cr^{3+}/Cr} = -0.74$ and $E^0_{Mn^{2+}/Mn} = -1.185$) among the five constituents, hence dominate early ($t \approx 0$) oxidation (Fig. 1b). Atomic radius and CED relate to atomic diffusivity, a property orthogonal to $E^0$ in determining the propensity to oxidize. Despite Mn's greater tendency to oxidize (lower $E^0$), Cr's smaller atomic size ($r_{Cr} = 189$ pm vs $r_{Mn} = 197$ pm) and lower CED ($CED_{Cr} = 8.24$ Pa vs $CED_{Mn} = 11.6$ Pa) make Cr more thermodynamically and kinetically favored to oxidize first, thus dominating the formation of the initial oxide layer (Fig. 1c). As oxidation continues, Ni, positioned centrally within the PIP relative to the other elements, acts as a phase stabilizer due to its high energetic barrier to structural change ($E_{BCC} \gg E_{FCC}$), making it the less likely to oxidize, hence its enrichment beneath the oxides. Associated stress, however, manifests as perturbations to Ni lattice parameters (see detailed discussion on associated x-ray below). Subsequently, slower oxidizing elements like Fe and Co start to diffuse towards the surface (Fig. 1d). After 30 min, oxidation shifts beyond the Cabrera-Mott (CM)[19] and into the Xu-Rosso-Bruemmer (XRB)[25] regime due to increased diffusion distance. As the oxide gets thicker, Fe and Co begin to dominate the outermost surface whilst Cr and Mn cease further movement and are enveloped to form the inner (buried) oxide layer (Fig. 1e). On extended reaction times, the composition of the surface oxide is inversely correlated to predictions of the PIP, due to the so-called Thermo-Oxidative Composition Inversion (TOCI, Fig. 1f)[31–33]. These later-stage TOCI oxides form by the release of the metal through fractals formed on the previously formed (inner) layer, associated elements oxides form as localized agglomerates or islands. The use of APT further illuminates' details about the growth of outer oxides, especially the formation of Co islands on the surface layer, which might not be visible in two dimensions (Fig. 1g). This work introduces an interactive method to predict oxidation behavior using PIP and applies APT and XANES to visualize elemental distribution and speciation during oxidation. It also maps the interactions between each element and diffusing oxygen with speciation and organization, revealing a reaction timeline (history), hence the underlying mechanism.

Figure 2 shows a CoCrFeNiMn alloy constituting a single-phase FCC structure that is oxidized at 400 °C for 120 min. A relatively lower temperature compared to past studies is chosen for this study to suppress the phase transformation tendency in this super-saturated alloy. The thin oxide film (~20 nm) forms a compositionally layered structure. Starting from the equiatomic base alloy, a Ni-enriched layer is followed by Cr-rich oxide followed by Fe, Mn, and Co-rich oxide layers (Fig. 1g). The enrichment of elements in the top layer, however, is not uniform. While the Mn layer appears on the top, at certain locations, bubble-like Fe pockets can be seen on the top layer of the surface (Fig. 2b). Furthermore, due to the solid-state nature of the oxidation, atomic diffusion from the bulk alloy results in the formation of defects within inner layers (Fig. 2c). Unlike liquids, solid surfaces undergo elastic deformation as atoms diffuse through to form the oxide layer. The inability to replace dislodged atoms (surface elasticity) leads to the formation of vacancies and defects, especially within the enriched/depletion regions where most of the atomic transfer happens (see supporting information for more details, Fig. S2). The

composition profile shows rather averaged changes on traversing from base alloy to outer later due to low statistics, yet the compositionally distinct layers (which is), Ni-rich, Cr-rich, Fe/Co-rich, and Mn-rich are very evident. A two-dimensional view of this system, however, does not provide the complete picture of the complex speciation of the oxide since the outer oxide is non-uniform. An average line scan of the TEM film (Fig. 2c) shows the expected gradient for the inner oxide but does not display the outer oxide enrichments (for Fe and Co) that are very apparent from the elemental map (Fig. 2d). Discrepancy between elements within the bulk alloy and especially in the enriched interface is due to migration of some elements (predominantly Mn) towards the surface layer. As more Mn cations diffuse out, more bulk atoms will become depleted and manifest themselves in the outermost surface layer.

To conduct a comprehensive evaluation and examine the structural modifications in the oxide film, circular averages of grazing incidence wide angle X-ray scattering (GIWAXS) data for oxidized samples are presented (see Figs. S3, S4, methods, and Fig. 2e, f). The lower q values reveal the presence of the Spinel structure (associated with Fe/Mn/Co oxide) and Corundum structure (corresponding to $Cr_2O_3$) as the outermost layer of the oxide, aligning with our TEM observations. Apart from detailing the oxidized layer's structure, noteworthy insights into the enriched layer and matrix are also provided (Fig. 2f).

The peak, corresponding to FCC (111) of the Ni-rich layer, shifts from q = 3.1 to 3.06 Å⁻¹ (2Θ shift from 44.7° to 44°) with an increasing incidence angle, indicating a progressive increase in lattice parameter from the top to the bottom region of the Ni-rich layer. Additionally, another FCC peak (at 3.01 Å⁻¹ or -2Θ = 43.3°) starts emerging at a deeper grazing angle, corresponding to the matrix (see the unoxidized sample in the inset) This observation suggests the formation of an additional FCC phase with a distinct lattice parameter between the oxide and base interface. Notably, this finding aligns with our TEM results illustrating a reoriented interface (Fig. S2).

Now hereon, APT becomes a powerful resource to capture a more complete oxidation profile across the whole surface. Employing the in-situ APT technique, a closer examination of oxidation-driven evolution was conducted. After subjecting the material to oxidation at 400 °C for 30 min (APT results for 2 min and 120 min are provided in the supplementary information), the resulting oxide layer exhibited clear inner and outer regions, accompanied by an enrichment layer beneath oxides and atop the base alloy. In-depth analysis of the 3D element distribution maps in Fig. 3a–c revealed intriguing details. The oxide in proximity to the oxide/metal interface, is characterized by a 28 at. % O iso-concentration surface (iso-surface) showcased a region enriched in Cr. Moving away from the oxide/metal interface, the outer oxide exhibited a relatively uniform distribution of Ni, albeit at a lower concentration than the enriched layer. Notably, this outer oxide displayed alternating areas enriched in Fe and Co, with discrete Co-enriched islands exhibiting reduced Fe content. Additionally, a faint enrichment of Mn was detected in the outer regions of the reconstructed volume. This 3D distribution offered valuable insights into the non-uniformities during the oxidation process, providing information not discernible through 2D films.

Semi-quantitative trends in elemental distribution are depicted through the 2D contour plots presented in Fig. 3d. This approach, involving examination of elemental distributions within a two-nanometer thick slice of interest at the specimen's center, allowed for enhanced visualization of distributions across the oxide/metal interface. These 2D slices in Fig. 3d from the reconstructions facilitate a clear differentiation of dominant regions, such as Cr-rich and Fe-rich zones, enabling the distinction between inner and outer oxide regions (refer to Figs. S5–S7 for further details). Furthermore, the region enriched in Ni shed light on the enrichment zone situated just beneath the oxide layer, prior to reaching the base alloy. The spatial

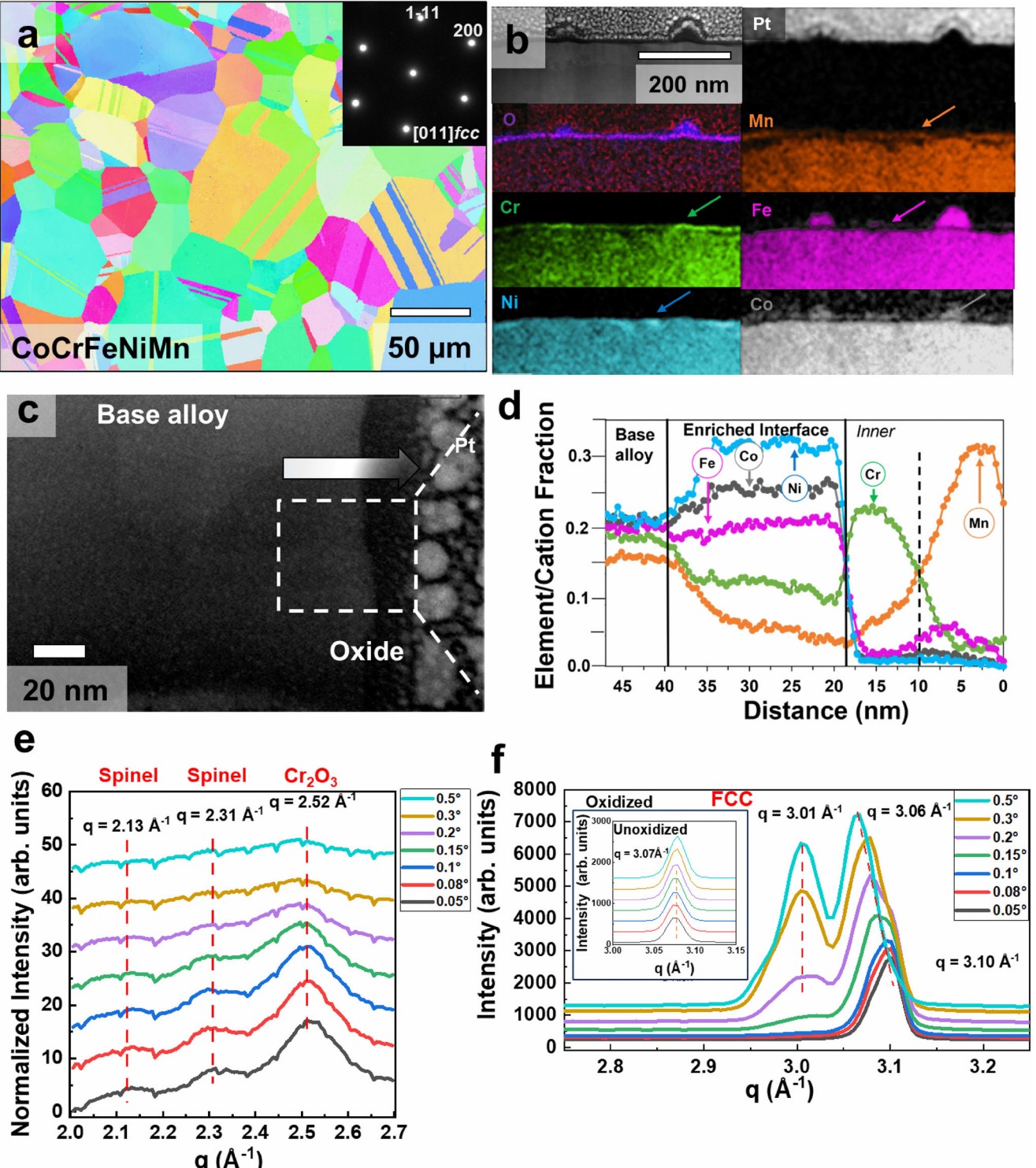

**Fig. 2 | Ex situ analysis of oxidized HEA (CoCrFeNiMn). a** Inverse pole figure map showing the randomly oriented recrystallized microstructure of the starting alloy condition. The inset shows the selected area diffraction pattern, further confirming the FCC structure and single phase. **b** A cross-section of the sample extracted using FIB after oxidation at 400 °C for 120 min at 10 mbar pressure inside the custom-made atom probe reactor chamber. The figure shows the STEM image and the elemental distribution in the oxide film and the base metal. **c** STEM image further magnifying the oxide film where an arrow is used to traverse from the base alloys to the oxide film. **d** TEM EDS linear profile of elemental distribution on traversing from base metal to oxide film. **e** GIWAXS traces from q = 2–2.7 Å⁻¹ and **f** from 2.75–3.25 Å⁻¹ with incidence angle varying from 0.05° (lower) to 0.5° (top). The lowest angle grazes the topmost oxide layer showing the presence of a Spinel structure and Corundum structure and as the grazing angle increases these phases disappear. Peak shift and peak splitting can be seen in (**f**) due to compositional changes in the matrix near the oxide metal interface.

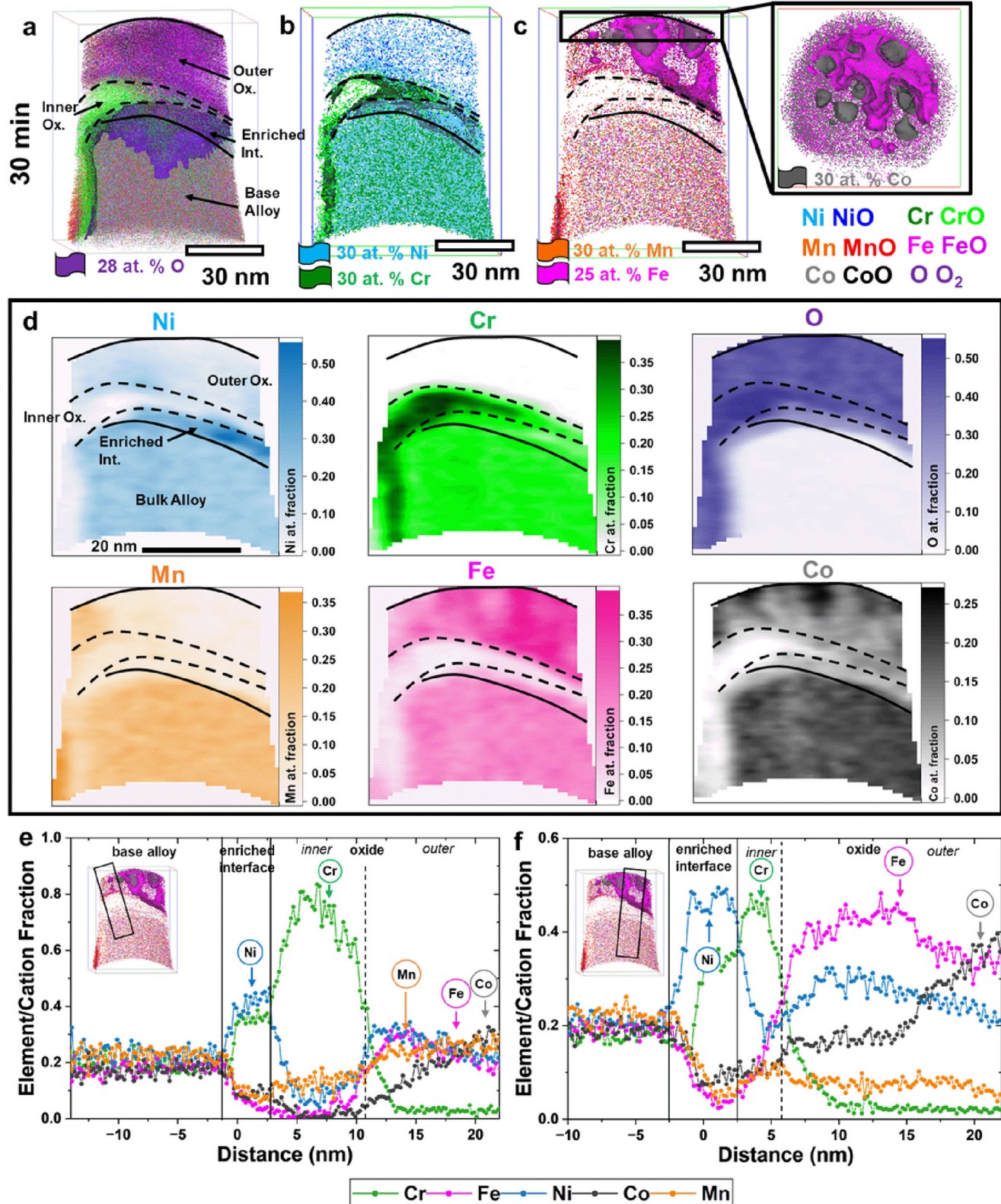

**Fig. 3 | APT data of CoCrFeNiMn alloy post-oxidation at 400 °C for 30 min. a** 3D reconstruction of the needle developed using interactive visualization APT software, all metal species are made visible here, and a purple-colored iso-concentration surface of oxygen is used to delineate the oxide from base metal. **b** Ni and Cr distribution within the needle is shown along with the iso-surfaces of Ni (cyan) and Cr (green) to demark the regions richer in these elements. **c** Separated Mn, Fe, and Co distribution along with the iso-surfaces of Mn and Fe (insert top-down view showing the sparse distribution of Co. **d** Elemental map of the overall sample. **e** Proxigram elemental distribution of sample taken from the indicated region in the insert. **f** Proxigram of a sample taken from different spaces within the sample.

distribution of Mn indicates varying degrees of dominance in certain outer regions, notably at the top-left and bottom-left corners (refer to Fig. 3d).

Concentration profiles for distinct regions of the reconstructed volume provide further insights. When analyzing the top-left corner (see Fig. 3e), the outer oxide is enriched in Mn, accompanied by Fe and Co. The depth of the inner oxide is not uniformly consistent across the sample. Within this region, the Cr-rich layer extends to approximately 8 nanometers before transitioning to a Ni-enriched zone (indicating a depletion of other elements). The proximity of Ni enrichment (-3 nm) to this Cr suboxide layer suggests Ni's entrapment at the metal/oxide interface, a result of crystallographic stabilization due to its role in stabilizing the FCC phase of the base metal. In the outer oxide, there's a depletion of Cr, giving way to regions predominantly dominated by Mn, Fe, and Co.

Conversely, when analyzing the top right corner (as depicted in Fig. 3f), where Co-rich islands are found, a significant variation emerges. In this region, the prevalence of Co and Fe is notably pronounced in the outer oxide, while Mn appears less abundant. Intriguingly, there's a substantial discrepancy in the depth of the inner oxide and enrichment region when compared to the orthogonal corner (Fig. 3e). The Cr-rich inner oxide spans -3 nm, whereas the Ni-rich interface extends to around 5 nm. This non-uniformity underscores the significance of examining oxidation from various angles and dimensions. We infer that given inherent surface elasticity, early oxidizing, and late oxidizing elements follow orthogonal diffusion paths and hence distribute differently on the surface. Despite the predictability of trends, controlling the direction of diffusion kinetics presents a more intricate challenge.

These APT results provide evidence that PIP (Fig. 1a) can provide a degree of predictability for the oxidation and speciation process for an otherwise highly entropic (and chaotic) system. Each element is speciated based on a combination of their affinity to oxygen as well as their diffusivity. As captured in the distribution of Cr (relatively small) Ni (structurally stabilized), predictions from PIP—a thermodynamic parameter, should be qualified with kinetic considerations in understanding the final organization and speciation. To further evaluate the oxidation mechanism, we introduced Al, a relatively small ($r_{Al} = 184$ pm vs $r_{Mn} = 197$ pm and $r_{Cr} = 189$ pm) lower $E^0$ element ($E^0_{Al^{3+}/Al} = -1.67$) to the equiatomic HEA. When 6 at. % of Al is added *in lieu* of Mn (reduced from 20 to 14 at. %), we observe distinct differences in the oxide layer for the same reaction conditions (Fig. 3). The microstructural properties of the modified alloy (before oxidation) where Mn was substituted partially by Al is provided in Fig. S8. Figure 4a shows Al propensity to surface partition based on PIP. The PIP predicts that Al, as the fastest oxidizing species, should create the first oxidation layer. The presence of Al, however, will have other consequences since additional interactions (with Al, specifically) can lead to colligative behavior, altering the propensity to diffuse and oxidize for other elements. From a two-dimensional film perspective, the alloy behaves as predicted by PIP and as previously shown with CoCrFeNiMn alloy, albeit with the addition of aluminum within the inner layer. Other elements are still ordered as expected. On extended oxidation, Co and Fe respectively dominate the outer oxides, followed by Mn, Cr, and then Al (Fig.4b–d (shown by the arrows) and Figs. S9, 10). The stability of Ni against oxidation is not perturbed by the addition of Al, hence it is enriched below the oxides. The APT result also displays a layered oxide; however, it appears more complex, with three apparent regions (Fig. 4c). First, we observe Al enrichment at the oxide/metal interface, adjacent to a region of Cr-enrichment, which we refer to here as the inner oxide. Adjacent to the region of Cr-enrichment, Mn and Co enrichment in the middle oxide layer is observed, and lastly, in the outermost oxide, we observe significant Fe-enrichment up to 40 at. % (Fig. 4d).

Beyond adding an extra layer to the complex oxide, the addition of Al suggests perturbations due to interaction between some elements even before oxidation happens. X-ray absorption near edge spectroscopy (XANES) results capture these interactions before and after oxidation and create a clear picture of the oxidation states for each element[34]. Figure 4e–i highlights these interactions as indicated by peak shifts, or lack thereof, before and after oxidation. Highly oxidizing elements such as Cr, Mn, and Fe have a high degree of interaction with Al, hypothetically since Al became the fastest oxidizing element and thus, the aluminum oxide layer is formed first.

The Al oxide layer becomes a diffusion barrier to any other element(s) that is being oxidized, akin to colligative property (vapor pressure depression), as Al lowers the propensity of other elements to diffuse towards the oxidant. The Cr L-edge spectrum (Fig. 4e) shows a minor shift with the addition of Al to the HEA. Furthermore, the small change between unoxidized and oxidized states indicates that $Cr_2O_3$ forms rapidly upon exposure, indicating the likely immediate concomitant formation of both $Al_2O_3$ and $Cr_2O_3$ oxides[35]. Mn L-edge spectrum (Fig. 4f), however, shows a change in the unoxidized condition with the addition of Al, where the oxidation state gets reduced to a lower state ($Mn^{2+}$, forming MnO). This suggests a strong interaction between Al and Mn, as without Al, Mn initially oxidizes to $Mn_2O_3$ ($Mn^{3+}$)[36,37], upon further oxidation, both alloys form the same oxide, $MnO_2$ ($Mn^{4+}$). The slower diffusing Fe shows a similar behavior to Mn where Al has an effect on to unoxidized state (Fig. 4g). Peaks shift from what was previously $Fe_3O_4$ ($Fe^{2+}/Fe^{3+}$) to $Fe_2O_3$ ($Fe^{3+}$) again showing the strong interaction between Fe and Al[38].

Al seems to have a minor effect on both Ni and Co, both of which are the weakest oxidizing component and it seems to have no effects on both unoxidized and oxidized states. We infer that Al addition does not have significant electronic interaction with Co and Ni (Fig. 4h, i), this is shown as their oxidation behavior remains the same in Fig. 4d. Figure 4j summarizes these interactions where Mn-Al tends to have the strongest interaction and both Ni and Co being the weakest.

This study uncovers the oxidation process in compositionally complex alloys. Our study involved subjecting an equiatomic CoCrFeNiMn alloy to oxidation at 400 °C under 10 mbar oxygen pressure for both ex situ and in situ observations. Ex situ observations, conducted using TEM after 120 min of oxidation, revealed a layered oxide structure. The 20 nm oxide layer on the equiatomic base alloy exhibited a composition sequence of Ni-rich (bottom layer), Cr-rich, and then Fe, Mn, and Co-rich layers (outer oxide). We chose a relatively low oxidation temperature to prevent phase transformation in the alloy, focusing solely on examining the oxidation process in the supersaturated alloy condition. The time-series observations at 400 °C unveiled a layered progression of oxidation. Controlled oxidation demonstrated the rapid formation of compositional layers, even after just 2 min of oxidation—starting with a Ni-enriched pre-oxide layer, followed by $Cr_2O_3$, and a multi-oxide layer rich in Fe, Co, and Mn. The Ni-enriched layer maintained an FCC structure, while $Cr_2O_3$ exhibited a corundum structure (trigonal $\bar{R}3c$ space), and the Fe-Mn-Co layer showed a Spinel structure. The layer formation follows a selective speciation process influenced by a combination of thermodynamics and kinetics, as predicted by PIP. Due to competition among speciating species, a single-phase Spinel-based complex oxide could have initially formed, subsequently decomposing into Fe-rich, Co-rich, and Mn-rich oxides. The interplay of various factors, such as standard reduction potential, cohesive energy density, and atomic radius, has been explored to predict and understand the oxidation sequence of different elements within the alloy. Using in-situ APT and XANES, the study tracks the oxidation process, highlighting interactions among elements during oxidation. Additionally, we examined an alloy with added aluminum (Al) and observed altered oxidation patterns and interactions among elements. The research enhances the

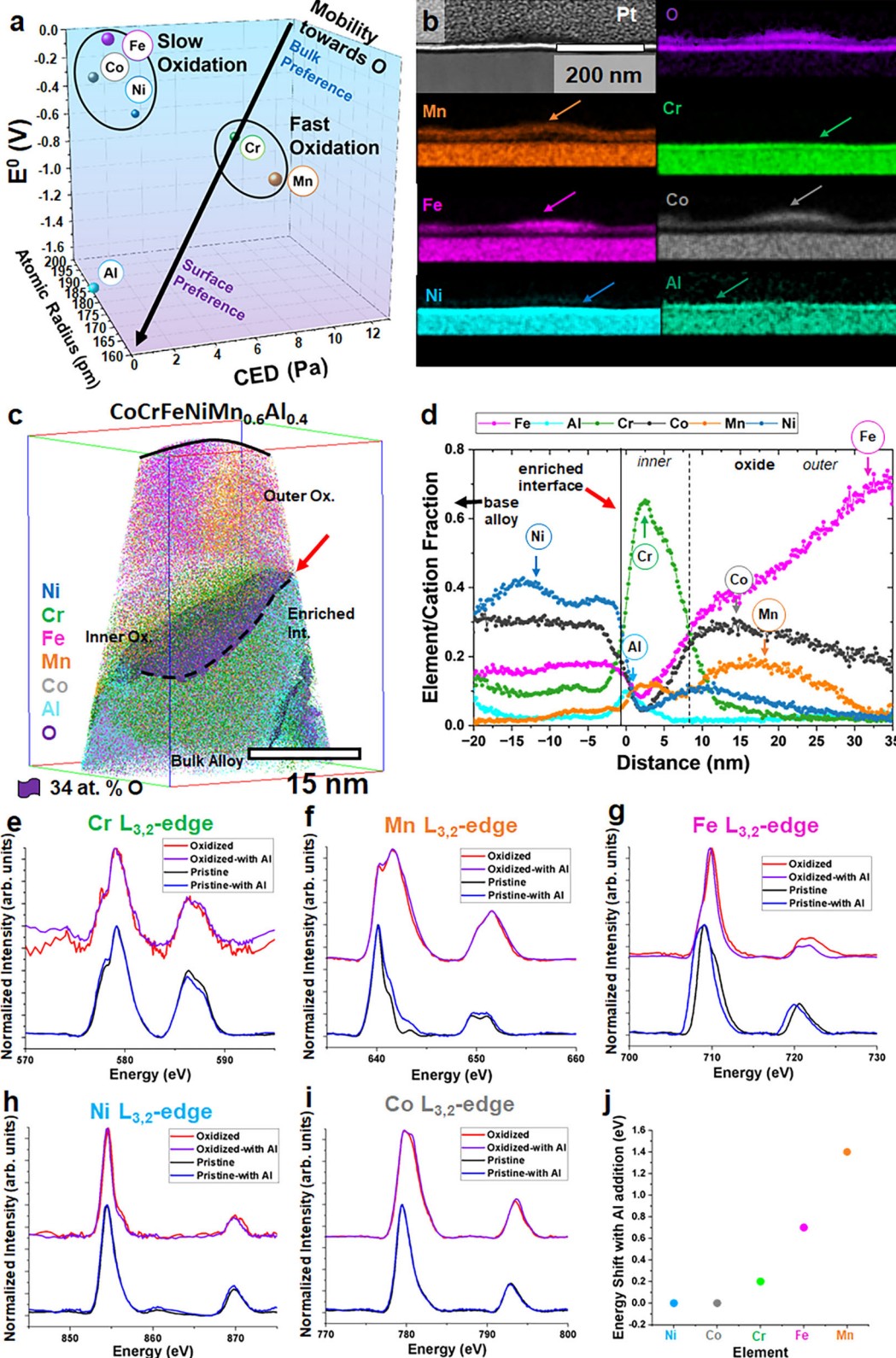

**Fig. 4 | Analysis of HEA with the addition of aluminum (CoCrFeNiMnO.6Al0.4).** **a** PIP for the HEA with the addition of Al. **b** TEM EDS map of thin film CoCrFeNiMn0.6Al0.4 sample **c** APT distribution of the HEA sample. **d** Proxigram of the interface indicated by the arrow in 4c. XANES spectra of **e** Cr L3,2, **f** Mn L3,2, **g** Fe L3,2, **h** Ni L3,2, **i** Co L3,2 for both CoCrFeNiMn and CoCrFeNiMn$_{0.6}$Al$_{0.4}$ alloys before and after oxidation. **j** Amount of shift with the presence of Al within the alloy.

understanding of oxidation in MPEAs/HEAs, facilitating alloy design for improved performance and durability.

## Methods

### Materials and experimental approach

The two alloy compositions (CoCrFeNiMn and CoCrFeNiMn+Al) (measured in atomic ratios) were synthesized using arc melting and casting techniques. Afterward, a homogenization annealing process was carried out at 1200 °C for a duration of 10 h. The alloys were then cold rolled to a 50% reduction in the cross-sectional area. Subsequently, recrystallization was induced via annealing at 1100 °C for 10 min, resulting in the formation of 3 mm thick sheets with refined microstructures. The alloy sheets were then precisely sectioned, mounted, and subjected to metallographic polishing.

### Microstructural characterization of the alloy before oxidation

For the initial microstructural analysis of the alloys, an Oxford Symmetry electron backscatter diffraction.

(EBSD) detector was employed. This EBSD system was integrated into a Thermo Fisher Scientific plasma-focused ion beam (PFIB). The obtained EBSD.

Data were subsequently analyzed using the integrated AZtecHKL software.

### Ex situ oxidation and characterization

Mirror-polished samples of the two alloys underwent oxidation within the chemical reactor chamber attached to the atom probe instrument. The flat specimens were securely held in place by a custom-made clip situated within the reactor chamber. In order to validate the oxide layer structures formed within the bulk samples, a metallographically polished bulk sample was subjected to an oxidation process at 10 mbar $O_2$ for a duration of 120 min at 400 °C. This procedure ensured consistency with the oxidation conditions applied to the in-situ samples. Subsequently, cross-sectional TEM samples were extracted from the oxidized surfaces using a site-specific focused ion beam (FIB) lift-out technique. The schematic of the experimental procedure for coupled ex situ and in situ characterization of oxide film formation on HEA surfaces is shown in Fig. S11. The manuscript contains scanning transmission electron microscopy−energy dispersive spectroscopy (STEM-EDS) maps derived from the cross-sectional TEM sample of the bulk alloy oxidized for 120 min.

### Grazing incident wide-angle X-ray scattering (GIWAXS)

GIWAXS measurements were performed at the Complex Materials Scattering (11-BM CMS) beamline at the National Synchrotron Light Source II (NSLS II), Brookhaven National Lab. The flat bulk samples that were within the reaction chamber attached to APT were used for the GIWAXS experiments. A schematic of the ex situ GIWAXs method is shown in Fig. S12. The GIWAXs measurements were performed with an incident X-ray beam with a 17.0 keV energy, and an incident angle varied from 0.05–0.5°. The beam size was 200 μm × 500 μm. The scattering signal was collected by an area detector, Pilatus 800 K.

### Cross-sectional STEM-EDS

TEM specimens were prepared using a Thermo Fisher Scientific Helios 5 Hydra DualBeam PFIB/SEM and an FEI Helios Nanolab FIB/SEM. STEM imaging and spectroscopy were performed using a Thermo Fisher Scientific Themis Z 60-300 S/TEM with an accelerating voltage of 300 kV and a convergence angle of 25 mrad. The high-resolution HAADF STEM images were collected with an annular detection range of 51–200 mrad. Twenty-five frames of the image with a dwell time of 4 μs per pixel were post-processed using a drift-corrected frame integration method in Velox software to optimize the signal-to-noise ratio and contrast. EDS was collected to show the compositional map using a Super-X G2 detector with a dwell time of 3 μs per pixel and 540

frames. EDS line profile was plotted perpendicular to metal–oxide interfaces and integrated over 50 pixels (3.5 nm). A part of STEM work was also performed using an aberration (CS) corrected, JEOL ARM200CF operated at 200 keV equipped with a JEOL Centurio energy dispersive spectrometer (EDS) with a 0.9 sR collection angle. Annular dark-field (ADF) data were collected with a convergence angle of 20 mrad and a collection angle ranging from 79–294 mrad. Digital imaging data collection was performed using Gatan Microscopy Suite (GMS) version 3.

### Atom probe tomography

APT sample preparation was executed using an FEI Helios dual-beam focused ion beam-scanning electron microscope (FIB-SEM). The process began with the deposition of a protective Pt capping layer, serving to shield the material from Ga ion-induced damage during FIB milling. A cantilever was extracted from the base alloy, and segments of it were affixed onto a commercial Si micropost array. These segments were then shaped into APT needle specimens.

The in situ APT experimental methodology encompasses several steps[39]. Initially, native oxide layers are analyzed and removed via APT field evaporation. Subsequently, the specimens are transferred under ultra-high vacuum (UHV) conditions ranging from $10^{-7}$ to $10^{-8}$ mbar to a gas-phase chemical reactor chamber preheated to the designated temperature of 400 °C. Within this chamber, the specimens are exposed to $O_2$ gas (99.993% purity) at 10 mbar for select time steps. Following the oxidation exposure, the APT specimens are transferred back to the analysis chamber under UHV conditions. The newly formed oxide and underlying base metal are then analyzed via APT. The oxidation process for the needle specimens is carried out at 400 °C for varying durations of 2, 5, 30, and 120 min. These parameters are chosen to induce the development of nanoscale oxide layers with thicknesses ranging from approximately 5 to 50 nm.

For APT data collection, a CAMECA Local Electrode Atom Probe (LEAP) 4000X HR system equipped with a 355 nm wavelength ultraviolet (UV) laser was employed. The user-defined settings included a laser energy of 200 pJ/pulse, a pulse repetition rate of 200 kHz, a specimen base temperature of 45 K, and a detection rate of 0.003 ions/pulse. The analysis chamber was maintained at a pressure below $2 \times 10^{-11}$ Torr. The detector efficiency of the LEAP utilized in this study is estimated to be around 36%. The collected data underwent reconstruction and analysis using IVAS version 3.8.2 by CAMECA.

### STEM of needle specimen after in situ oxidation

Crystallographic and compositional examination of oxidized needle samples was done with a probe-corrected FEI Titan 80–300 STEM instrument operated at 300 kV. The observations were performed using STEM with a high-angle annular dark-field (HAADF) detector. The probe convergence angle was 18 mrad, and the inner detection angle on the HAADF detector was three times larger than the probe convergence angle. AZtec energy dispersive spectroscopy (EDS) software was used to analyze the composition captured by EDS in the STEM instrument.

### X-ray absorption near edge spectroscopy (XANES)

The electronic structure changes of unoxidized CoCrFeNiMn and CoCrFeNiMn+Al alloys, and samples of each alloy oxidized at 400 °C for 2 min, 30 min, and 120 min, and Co metal, $Cr_2O_3$, Fe metal, $Fe_2O_3$, $Fe_3O_4$, and $MnO_2$ standards were analyzed using synchrotron-based XANES. XANES was performed at the Advanced Light Source (ALS) synchrotron beamline 7.3.1 at Lawrence Berkeley National Laboratory (LBNL). Experiments were conducted in an Ultra-High Vacuum chamber with a background pressure of $5.0 \times 10^{-8}$ Torr with a ring current of 500 mA. XANES data were collected at room temperature using Total Electron Yield (TEY) mode, which measures the sample drain current resulting from photo and Auger electrons leaving the sample surface.

The incident beam intensity was monitored ($I_o$) via the gold grid to normalize the total electron yield signal ($I_t$). Each final spectrum was an average of four scans. For comparison, asymmetric least squares smoothing baseline was performed and then normalized. The Co, Cr, Fe, Ni, and Mn L-edge TEY signals were utilized in this work to characterize the top 10 nm of all samples to complement APT and TEM studies.

## Data availability

All data were available in the main text or the supplementary materials.

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

## Acknowledgements

A part of the experimental work done at Pacific Northwest National Laboratory (PNNL) for this research was supported by the Department of Energy (DOE), Office of Science, Basic Energy Sciences, Materials

Sciences, and Engineering Division as a part of the Early Career Research Program FWP 76052. PNNL is operated for the US DOE by Battelle Memorial Institute under Contract No. DE-AC05-76RLO1830. The XANES analysis was conducted at beamline 7.3.1 at the Advanced Light Source at Lawrence Berkeley National Laboratory, a national user facility. The beamline is supported by the Director, Office of Science, Office of Basic Energy Sciences of the US DOE. Another part of this research was supported by ONR under grant N00014-23-1-2758 at NC State University.

## Author contributions

Conceptualization, project administration, funding acquisition: B.G. and A.D.; Methodology: B.G., S.L., and E.K., Formal analysis: B.G., A.Mar, E.K., M.O., R.L., MT., and A.Mal, Data curation: B.Guo, A.Mar, A.K.B., T.S., F.Y., J.G., and A.A. Writing—original draft: B.G., A.Mar, and E.K.; Writing—review and editing: M.T., A.D., A.A.

## Competing interests

The authors declare no competing interests.
