## [Peer Review File · Nature Communications]

Mechanistic Understanding of Speciated Oxide Growth in High Entropy AlloysREVIEWER COMMENTS

Reviewer #1 (Remarks to the Author):

This study systematically investigates the oxidation mechanisms of equiatomic CoCrFeNiMn and CoCrFeNiMn_{0.6}Al_{0.4} MPEAs through a comprehensive experimental study. Advanced materials characterization techniques—namely, in situ atom probe tomography (APT), transmission electron microscopy (TEM), and X-ray absorption near-edge structure (XANES)—were employed to delve into the oxidation process. The investigation encompassed varied oxidation durations of 2, 30, and 120 minutes, focusing on the evolution of oxide composition, particularly the specialized growth of oxides for each principal element. The findings reveal distinct oxidation kinetics among the elements. Notably, Cr and Mn exhibit relatively rapid migration, reaching the outermost surface in the initial stages of oxidation (2 and 30 minutes), whereas Co and Fe predominate in the outer oxide layer after 120 minutes, displacing the former elements. This dynamic evolution underscores the complexity of oxidation behaviors within these materials. An experimental breakthrough lies in the application of in situ APT, allowing for precise observation of surface oxide compositional changes during the early oxidation phases. The introduction of the 'preferential interactivity parameter (PIP)' stands as a theoretical innovation proposed by the authors. This parameter offers a predictive measure of the oxidation propensity for each principal element, indicating their inclination either toward surface migration or retention within the bulk.

Overall, this study furnishes compelling experimental evidence delineating element speciation during MPEA oxidation. Moreover, the novel concept of PIP introduces a pragmatic metric guiding the design of MPEAs for tailored oxidation resistance.

Below are several questions that should be addressed before I can recommend publication.

1. The formation of oxide phases remains unexplored. Is the spatial distribution influenced by the solubility of oxides in each other? Essentially, does the layering result from the separation of oxide phases? A comprehensive X-ray diffraction (XRD) analysis, potentially employing grazing incidence XRD, could provide valuable insights by precisely characterizing the phase formation concerning the thickness of oxide layers. While certain phase indexing has been conducted using TEM, it's essential to note that TEM probes only a minute portion of the surface. Such results can be then compared to phase diagram calculations to see if the predicted oxides are at equilibrium or far from it, especially in the early oxidation stage.
2. The introduction of PIP is a novel theoretical development. However, it is a rather simplified term, whose limitations should be clearly discussed. Would it work for some simple binary or ternary concentrated systems? This validation could be done based on previous literature results. For example, in a Ni-Al alloy system, the simultaneous growth of more than one oxide such as a Al₂O₃ and NiO occur, with different growth rates. In addition, there could be competition among the diffusion and reaction processes. This term also ignores the thermodynamic database rather than relying on simplifying assumptions for ideal solution models.
3. In fig 2d, why is Mn concentration in the bulk alloy lower than other elements?
4. After 2 hrs oxidation, why would the outmost surface oxide become enriched in Co and Fe, which are predicted to have slow oxidation per fig 1a? All other elements seem to agree with the PIP predictions at 2 hrs.
5. Is the sample in fig 1g the same as the one in fig 2d? Why are the elemental profiles different?
6. The sequence of results presented are somewhat confusing. It might be easier for readers to follow to talk about results of 2 min, 30 min, and lastly, 2 hr oxidation. Right now, 2 hr oxidation is discussed

before the 30 min oxidation results.

7. Is it possible to calculate PIP for more metallic elements in the periodic table, so that readers can easily refer to this plot for future alloy design?

Reviewer #2 (Remarks to the Author):

Understanding the oxide composition during the oxidation or passivation of HEAs is of great interest both for fundamental research of the scientific community and for industrial application. The study presented here is valuable for the field and reveals the composition of in situ-grown oxide film on the surface of equiatomic CoCrFeNiMn alloys.

However, I have a couple of minor comments and 1 major concern.

Major: As one of the parameters for PIP prediction the authors used the standard redox potential. I believe that this parameter can't be used here, as the standard redox potential is for the standard conditions while 400C was used in the study, and, what is more important, it is only for aqueous systems. While Nernst potential (non-standard redox) can be used in the non-aqueous but any other liquids, any of the (standard or Nernst) can not be used in the oxidation in air/oxygen prediction. I was confused about why formation energies were used for the prediction instead. The formation energies could be found or calculated close enough for the temperature of the interest - 400C.

Minor: 1. Even though I hope it disappears during the revision, I should mention that all the potentials in the paper are missing units that are V(SHE), and redox couples are not written. Ecr is a poor labeling, should be $E_0(\text{Cr}^{3+}/\text{Cr})$ as the value was used for this redox couple, etc.
2. In the introduction authors mentioned the importance of these alloys and oxidation experiments for corrosion, however, didn't mention any studies that were done in the corrosion field. Please, add the paragraph showing the progress in HEAs passivation experiments, some of them were done employing APT as post-mortem analysis of grown film as well.

In general, the study is well-organized and helpful for the field. However, the potential issue should be resolved.

Reviewer #3 (Remarks to the Author):

This paper studies the passive film formed on a typical high entropy alloy surface. A parameter is introduced to understand the thermodynamic tendency and also a tool to understand the oxidation behaviours. A few advanced techniques were employed such as APT, TEM. I think that it is an interesting paper and the authors have carried out some in-depth discussion. However, it still lacks of more in-depth discussion, such as the interaction between various elements. It is a really complex issue and the authors should have sought more advanced analysis. The oxidation is a thermodynamic tendency, but the oxidation process is more kinetic. It is not easy and it requires more tests to overview the kinetic process.

Response to Reviewer Comments (NCOMMS-23-52662-T)

The authors express gratitude for the thoughtful review and valuable feedback on our manuscript entitled "Mechanistic Understanding of Speciated Oxide Growth in High Entropy Alloys". The additional insights and suggestions provided have been instrumental in improving the quality of our work, and we appreciate the time and effort you dedicated to the review process.

In this document, we address each comment provided by the reviewers. The revisions incorporating these comments have been implemented in the updated manuscript and are indicated by yellow highlights.

Reviewer #1

Q1.1. The formation of oxide phases remains unexplored. Is the spatial distribution influenced by the solubility of oxides in each other? Essentially, does the layering result from the separation of oxide phases? A comprehensive X-ray diffraction (XRD) analysis, potentially employing grazing incidence XRD, could provide valuable insights by precisely characterizing the phase formation concerning the thickness of oxide layers. While certain phase indexing has been conducted using TEM, it's essential to note that TEM probes only a minute portion of the surface. Such results can be then compared to phase diagram calculations to see if the predicted oxides are at equilibrium or far from it, especially in the early oxidation stage.

A1.1. We value your feedback on the unexplored aspect of oxide phases in our research. To address this inquiry, we present a brief overview of our paper. Our study involved subjecting an equiatomic CoCrFeNiMn alloy to oxidation at 400°C under 10 mbar oxygen pressure for both ex-situ and in-situ observations. We chose a relatively low oxidation temperature to prevent phase transformation in the alloy, focusing solely on examining the oxidation process in the super-saturated alloy condition.

Ex situ observations, conducted using TEM after 120 minutes of oxidation, revealed a layered oxide structure. The 20 nm oxide layer on the equiatomic base alloy exhibited a composition sequence of Ni-rich (bottom layer), Cr-rich, and then Fe, Mn, and Co-rich layers (outer oxide) (refer to Figure 1g).

Despite these observations after 120 minutes, the crucial question arises: what transpired during the early stages of oxidation? As pointed out by the reviewer, is the spatial distribution influenced by the solubility of the oxides in each other, suggesting a complex oxide formation initially? To address these queries, we performed in situ experiments with time intervals ranging from 2 to 120 minutes, enabling a step-by-step observation of oxide growth.

The time-series observations at 400°C unveiled a layered progression of oxidation. For instance, Cr₂O₃ was the initial oxide to form, persisting even when the sample was transferred from a focused ion beam (FIB) to atom probe tomography (APT). Controlled oxidation demonstrated the rapid formation of compositional layers, even after just 2 minutes of oxidation - starting with a

Ni-enriched pre-oxide layer, followed by Cr_2O_3 , and a multi-oxide layer rich in Fe, Co, and Mn. The Ni-enriched layer maintained an FCC structure, while Cr_2O_3 exhibited a corundum structure (trigonal $\bar{R}3c$ space), and the Fe-Mn-Co layer showed a Spinel structure. The layer formation seemed to follow a selective speciation process influenced by a combination of thermodynamics and kinetics, as predicted by the PIP. Due to competition among speciating species, a single-phase Spinel-based complex oxide could have initially formed, subsequently decomposing into Fe-rich, Co-rich, and Mn-rich oxides.

To clarify the ideology behind the paper and the knowledge gap in the literature, the following has been added to the introduction:

“The structure and stability of HEAs surface oxides, however, are not well-established largely due to a wide variety of compositions and the limited understanding of the atomic scale mechanisms underlying the oxidation process. The oxide formation tendencies in a solution can thermodynamically be roughly deduced by Ellingham diagrams. However, considering many of these alloys are supersaturated-solid solutions, the concomitant oxidation and phase transformation at elevated temperatures are difficult to unveil^{2,8,9}. Application of such compositionally complex systems demands a multiscale analysis of the degradation mechanism underpinning their structural instability.”

The reviewer suggested conducting glancing incidence XRD and comparing the results with thermodynamic predictions. We appreciate that suggestion. We collaborated with National Synchrotron Light Source II, Brookhaven National Laboratories and were able to conduct grazing wide-angle X-ray scattering (GIWAXS) experiments. These experiments were performed around the critical angle of total external reflection allowing GIWAXS to probe the near-surface structure of the oxide at shallow angle and progressively deeper at higher angles. These experiments further confirm the presence of structurally different layers in the oxide film. Additionally, the results also reveal structural details regarding the change in structure and lattice parameters when transitioning from the Ni-enriched layer to the based metal.

The latest results are added to the revised manuscript and a summary is also provided below:

Fig. 2. Ex situ Analysis of oxidized HEA (CoCrFeNiMn). (a) inverse pole figure map showing the randomly oriented recrystallized microstructure of the starting alloy condition. The inset shows the selected area diffraction pattern further confirming the FCC structure and single phase (b) A cross-section of the sample extracted using FIB after oxidation at 400°C for 120 mins at 10 mbar pressure inside the custom-made atom probe reactor chamber. The figure shows the STEM image and the elemental distribution in the oxide film and the base metal (c) STEM image further magnifying the oxide film where an arrow is used to traverse from the base alloys to the oxide film. (d) TEM EDS linear profile of elemental distribution on traversing from base metal to oxide film. (e) GIWAXS traces from $q=2-2.7 \text{ \AA}^{-1}$ and (f) from $2.75-3.25 \text{ \AA}^{-1}$ with incidence angle varying from 0.05° (lower) to 0.5° (top). The lowest angle grazes the topmost oxide layer showing the presence of a Spinel structure and Corundum structure and as the grazing angle

increases these phases disappear. Peak shift and peak splitting can be seen in (f) due to compositional changes in the matrix near the oxide metal interface.

“To conduct a comprehensive evaluation and examine the structural modifications in the oxide film, circular averages of Grazing Incidence Wide Angle X-ray Scattering (GIWAXS) data for oxidized samples are presented (see Figure S3-S4, methods, and Fig. 2 (e-f)). The lower q values reveal the presence of the Spinel structure (associated with Fe/Mn/Co oxide) and Corundum structure (corresponding to Cr_2O_3) as the outermost layer of the oxide, aligning with our TEM observations. Apart from detailing the oxidized layer's structure, noteworthy insights into the enriched layer and matrix are also provided (Fig. 2f).

The peak, corresponding to FCC (111) of the Ni-rich layer, shifts from $q=3.1$ to 3.06 \AA^{-1} with an increasing incidence angle, indicating a progressive increase in lattice parameter from the top to the bottom region of the Ni-rich layer. Additionally, another FCC peak starts emerging at a deeper grazing angle (at 3.01 \AA^{-1}), corresponding to the matrix. This observation suggests the formation of a new FCC phase with a distinct lattice parameter between the oxide and base interface. Notably, this finding aligns with our TEM results illustrating a reoriented interface (Fig. S2).”

Furthermore, using Calphad for these predictions may not be accurate since most models in the database are tailored for high-temperature phase fields. At lower temperatures, where processes slow down, enabling the observation of various thermodynamically driven steps with spatial and temporal resolution, there is some correlation between oxide layering and thermodynamics. Yet, deviations from predictions suggest kinetic influences. To address these discrepancies, we developed a more sophisticated tool in the form of PIP.

Ellingham diagram, the standard free energies of oxide formation at 400°C rank as $\text{Al} < \text{Mn} < \text{Cr} < \text{Fe} < \text{Ni} \approx \text{Co}$.

The normalized PIP plot is displayed above. To emphasize, the advantages of PIP stem from its consideration not only of a single thermodynamic factor but of three distinct aspects:

External (E0): This denotes the tendency to exchange electrons with the surroundings, specifically with oxygen in this case. A lower value indicates a higher likelihood of electron exchange.

Internal (CED): Similar to binding energy, this factor represents the energy required to break away from the bulk. A lower value suggests easier detachment.

Diffusion (radius): This factor indicates the mobility of the diffusing atom, with D being inversely proportional to the radius. Smaller atoms exhibit higher mobility, making them easier and faster to diffuse.

The normalized plot provides a PIP scale that extends beyond High Entropy Alloys (HEA), encompassing all elements in the periodic table. For instance, examining only the E0 provides information akin to the Elingham diagram, where Mn's E0 scale is at 55 and Cr is at 62. However, when combining all three factors, Cr is positioned at 2.21, and Mn at 2.24.

Additionally, the normalized plot allows for the incorporation of multiplication factors for each of these aspects, providing flexibility based on the perceived importance of each factor. Currently, they are treated as equals.

According to the PIP values:

Al > Ni > Cr > Mn > Fe > Co

1.92 > 2.15 > 2.21 > 2.24 > 2.33 > 2.47

The anomaly observed in nickel is discussed in relation to structural/phase stability, which is currently not accounted for in the PIP.

Q1.2. The introduction of PIP is a novel theoretical development. However, it is a rather simplified term, whose limitations should be clearly discussed. Would it work for some simple binary or ternary concentrated systems? This validation could be done based on previous literature results. For example, in a Ni-Al alloy system, the simultaneous growth of more than one oxide such as a Al₂O₃ and NiO occur, with different growth rates. In addition, there could be competition among the diffusion and reaction processes. This term also ignores the thermodynamic database rather than relying on simplifying assumptions for ideal solution models.

A1.2. We appreciate the reviewer's sentiment towards PIP. We understand that PIP is very simplified as it is inherently a "thermodynamic" tool, where the kinetics are not put into consideration. We have recently explored some case studies for different alloy systems (from 2-5 components) and have discovered a good agreement with several literatures and what PIP predicts. There are also other major factors that play a role in the speciated ordering during oxidation such as temperature (affecting the diffusion and reaction as the reviewer mentioned),

atmospheric conditions (reducing/oxidizing), stoichiometry, compositional entropy, phase, and many more. These findings are summarized in a perspective.

Martin, Andrew, and Martin Thuo. "Predicting Emergence of Nanoscale Order in Surface Oxides through Preferential Interactivity Parameter." *ACS nano* (2024).

<https://pubs.acs.org/doi/abs/10.1021/acsnano.3c10935>

Q1.3. In fig 2d, why is Mn concentration in the bulk alloy lower than other elements?

A1.3. The TEM sample (2-hour condition) shows the time frame where Mn migration currently dominates as reflected by the surface domination of Mn oxide where this would be overtaken later by Fe and Co at an extended period of time. This is also spatially dependent on the location where the assessment is made. The lower Mn concentration within the bulk right below the Mn-rich outer oxide is accounted for by the surface oxide formation. As more Mn cations diffuse out towards the surface, more bulk atoms will become depleted (this is also reflected more strongly within the enriched interface, Fig. 2d).

We have added this clarification in the main text (p.4)

“Discrepancy between elements within the bulk alloy and especially in the enriched interface is caused due to migration of some elements (predominantly Mn) towards the surface layer. As more Mn cations diffuse out, more bulk atoms will become depleted and manifest themselves in the outermost surface layer.”

Q1.4. After 2 hrs oxidation, why would the outmost surface oxide become enriched in Co and Fe, which are predicted to have slow oxidation per fig 1a? All other elements seem to agree with the PIP predictions at 2 hrs.

A1.4. We infer that this is due to formation of some voids and cracks, especially within the enriched interface (see Figure S2) that creates new diffusion pathways that allow for slower diffusing atoms to “leak” out towards the surface, which is why they form islands, instead of

uniform layers. This behavior is similar to thermo-oxidative composition inversion (TOCI) seen in metal particles in our previous work where the surface speciation may become inverted once the surface oxide becomes thick enough to behave solid-like and thus fractures and allows for slower diffusing elements to come out (see refs 27-29, Cutinho, J. et al. <https://doi.org:10.1021/acsnano.8b01438>, 28, Martin, A. et al. <https://doi.org:10.1039/D0MH01832E> 29, and Martin, A., Kiarie, W., Chang, B. & Thuo, M. *Angew. Chem. Int. Ed.* 59, 352-357 (2020)).

We highlighted this in the following text in the introduction (p.3)

“In the long run, the oxidation pattern displays an inverse correlation with the bulk preferability across the oxide thickness, akin to the Thermo-Oxidative Composition Inversion (TOCI, Fig. 1f).²⁷⁻²⁹ The use of APT illuminates further details about the growth of outer oxides, especially the formation of Co islands on the surface layer, which might not be visible in two dimensions (Fig. 1g).”

Q1.5. Is the sample in fig 1g the same as the one in fig 2d? Why are the elemental profiles different?

A1.5. The outcomes depicted in Figure 1 stem from atom probe analysis utilizing conically shaped needle specimens, oxidized within the APT chamber. In contrast, Figure 2 illustrates findings from a bulk specimen where a flat sample underwent oxidation in a similar environment, and a TEM sample was extracted for post-oxidation characterization. Recognizing the intricate compositional changes inherent in the oxidation of multi-element systems, we posit that surface finish, morphology, and local structure can significantly alter compositions and reaction rates, particularly in the early stages.

Although the observed trends are consistent, the compositions captured by TEM differ from those obtained through in situ experiments in APT. The primary factor contributing to this variance seems to be the local geometry of the test piece. This aspect has been incorporated into the revised manuscript.

Q1.6. The sequence of results presented are somewhat confusing. It might be easier for readers to follow to talk about results of 2 min, 30 min, and lastly, 2 hr oxidation. Right now, 2 hr oxidation is discussed before the 30 min oxidation results.

A1.6. We regret any confusion, and the rationale behind the current structure of the manuscript is centered on our emphasis on showcasing APT as a comprehensive methodology for analyzing oxide growth. The inclusion of the TEM sample analysis (Figure 2) at an earlier stage is intentional, serving the purpose of illustrating that a TEM analysis alone may be considered incomplete. Our intention is to emphasize that a more comprehensive and nuanced understanding of oxide growth can be obtained by incorporating APT, providing a more holistic and detailed perspective.

Q1.7. Is it possible to calculate PIP for more metallic elements in the periodic table, so that readers can easily refer to this plot for future alloy design?

A1.7. Yes, absolutely. We are actually currently developing an application that allows users to generate PIP for any metals in the periodic table. The software is currently undergoing beta testing and will soon be made available. We are still currently awaiting approval from our funding agency before we can release the code.

Reviewer #2

Q2.1. As one of the parameters for PIP prediction the authors used the standard redox potential. I believe that this parameter can't be used here, as the standard redox potential is for the standard conditions while 400C was used in the study, and, what is more important, it is only for aqueous systems. While Nernst potential (non-standard redox) can be used in the non-aqueous but any other liquids, any of the (standard or Nernst) can not be used in the oxidation in air/oxygen prediction. I was confused about why formation energies were used for the prediction instead. The formation energies could be found or calculated close enough for the temperature of the interest - 400C.

A2.1. We understand the reviewer's concern towards PIP. PIP is inherently a thermodynamic prediction tool that takes these intrinsic properties at face value (during ground state), alas PIP's role is prediction during the earliest initial oxidation where the materials are still stochastically distributed, and no oxides are formed. As an example, this mixture of Cr-Mn-Fe-Co-Ni would be predicted to have a speciation of bulk>Co>Fe>Ni>Mn>Cr>surface initially, which is shown in the manuscript. However, at higher temperature, we agree with the reviewer that the kinetics would dominate and therefore goes beyond the realm of PIP. Especially with the complexity of other external effects affecting the reaction kinetics, this would definitely require higher dimensionality in the parameter. At the moment, PIP does not account for the reaction flux at the extended stage, just the initial condition.

Q2.2. Even though I hope it disappears during the revision, I should mention that all the potentials in the paper are missing units that are V(SHE), and redox couples are not written. Ecr is a poor labeling, should be $E_0(\text{Cr}^{3+}/\text{Cr})$ as the value was used for this redox couple, etc.

A2.2. We thank the reviewer for the suggestion and have since changed the notations in the revised manuscript.

Q2.3. In the introduction authors mentioned the importance of these alloys and oxidation experiments for corrosion, however, didn't mention any studies that were done in the corrosion field. Please, add the paragraph showing the progress in HEAs passivation experiments, some of them were done employing APT as post-mortem analysis of grown film as well.

A2.3. The introduction has been modified with the addition of literature and references highlighting the past work and existing knowledge gap.

“The structure and stability of HEAs surface oxides, however, are not well-established largely due to a wide variety of compositions and the limited understanding of the atomic scale mechanisms underlying the oxidation process. The oxide formation tendencies in a solution can thermodynamically be roughly deduced by Ellingham diagrams. However, considering many of these alloys are supersaturated-solid solutions, the concomitant oxidation and phase transformation at elevated temperatures are difficult to unveil^{2,8,9}. Application of such compositionally complex systems demands a multiscale analysis of the degradation mechanism underpinning their structural instability.

At temperatures exceeding 500°C, the oxidation behavior of traditional and complex alloys (e.g., Ni-Cr, Ni-based superalloys, Fe-Cr, CoCrFeNiMn 'Cantor' alloy) has been extensively studied in oxygen-rich environments and over extended periods (from days to hundreds of days^{5,10-20}). Key contributors to the process include species transport to the oxide/metal junction within the base alloy, movement across the oxide/metal interface, and passage through the oxide layer. For instance, atmospheric oxidation of CoCrFeNiMn-containing alloys results in a layered oxide film, with Cr-rich inner and Mn-rich outer oxides formed mainly due to outward cation diffusion¹⁴. In a study by Kim et al., the high-temperature oxidation of FeCrMnNiCo (900-1100°C in 20% O₂/80% N₂ for 24 hours) showed the formation of layered Cr₂O₃ and Mn₂O₃ oxides at 900°C. This process led to Ni, Fe, and Co enrichment in the base alloy. At 1000-1100°C, the outer Mn₂O₃ layer changed to Mn₃O₄, forming a (Mn, Cr)₃O₄ ternary oxide with Ni, Fe, and Co in solid solution. The presence of Mn, known for its high mobility and low-energy oxide formation, can deplete Mn in the alloy, causing pore formation and accelerating oxide growth.

Traditional knowledge suggests the chromium content in alloys is crucial for improving oxidation and corrosion resistance, forming a protective chromia layer. High-temperature materials use passivating elements like Al, Cr, or Si to form protective oxides. Cr's outward diffusion is faster than oxygen's inward diffusion, so chromia scales usually grow outward, consisting of fractions of iron, nickel, and manganese. However, at temperatures above 900°C, chromia can form volatile CrO₃. Aluminum, another effective alloying element, forms a protective Al₂O₃ layer, offering significant resistance to high-temperature oxidation and corrosion. Al₂O₃ scales grow slower than chromium oxide scales and are more stable, especially against aggressive carbon or sulfur species. However, the formation of alumina and consequent depletion of Al in the alloy could result in structural instability in the alloy []. The formation of oxide scales in HEAs containing multiple oxide forming species is influenced by their intricate chemistries and microstructures. High temperature, long-duration oxidation studies on bulk alloys typically reveal dominant oxide films composed of complex ternary and quaternary oxides. However, the polycrystalline nature of these alloys, combined with surface heterogeneity and phase instabilities at high temperatures, leads to highly variable oxide structures. These structures are the result of complex interactions. Bhargavi et al in their review summarize the observations made in different HEAs during high temperature oxidation, and not surprisingly it is difficult to develop a theoretical understanding or trends. Without independently examining each of these variables, a comprehensive

understanding of oxidation pathways and scale stabilities, and the subsequent ability to engineer their physical properties, could remain elusive.”

Reviewer #3

Q3.1. This paper studies the passive film formed on a typical high entropy alloy surface. A parameter is introduced to understand the thermodynamic tendency and also a tool to understand the oxidation behaviours. A few advanced techniques were employed such as APT, TEM. I think that it is an interesting paper and the authors have carried out some in-depth discussion. However, it still lacks of more in-depth discussion, such as the interaction between various elements. It is a really complex issue and the authors should have sought more advanced analysis. The oxidation is a thermodynamic tendency, but the oxidation process is more kinetic. It is not easy and it requires more tests to overview the kinetic process.

A3.1. We are grateful for your positive feedback and recognition. Your constructive comments regarding the need for a more thorough discussion, especially concerning the interaction between various elements in the alloy, are highly valued. Recognizing the complexity of the issue, we agree that additional analyses enriched the depth of our discussion. We appreciate your insights into the kinetic aspects of the oxidation process. Understanding the significance of kinetics, we have developed the PIP as a universal tool that extends beyond the Ellingham diagram, enhancing our ability to predict oxidation tendencies and understanding the nature of oxide films in complex alloys.

In response to the reviewer's suggestions, we have made several modifications, including an enhanced literature review, analysis, and additional experiments. These updates have been incorporated into the revised manuscript. Responses to Reviewer1 Q1 and Reviewer2 Q3 specifically also respond to current reviewer's concern and hence are provided below:

Our study involved subjecting an equiatomic CoCrFeNiMn alloy to oxidation at 400°C under 10 mbar oxygen pressure for both ex-situ and in-situ observations. We chose a relatively low oxidation temperature to prevent phase transformation in the alloy, focusing solely on examining the oxidation process in the super-saturated alloy condition.

Ex situ observations, conducted using TEM after 120 minutes of oxidation, revealed a layered oxide structure. The 20 nm oxide layer on the equiatomic base alloy exhibited a composition sequence of Ni-rich (bottom layer), Cr-rich, and then Fe, Mn, and Co-rich layers (outer oxide) (refer to Figure 1g).

Despite these observations after 120 minutes, the crucial question arises: what transpired during the early stages of oxidation? As pointed out by the reviewer, is the spatial distribution influenced

by the solubility of the oxides in each other, suggesting a complex oxide formation initially? To address these queries, we performed in situ experiments with time intervals ranging from 2 to 120 minutes, enabling a step-by-step observation of oxide growth.

The time-series observations at 400°C unveiled a layered progression of oxidation. For instance, Cr₂O₃ was the initial oxide to form, persisting even when the sample was transferred from a focused ion beam (FIB) to atom probe tomography (APT). Controlled oxidation demonstrated the rapid formation of compositional layers, even after just 2 minutes of oxidation - starting with a Ni-enriched pre-oxide layer, followed by Cr₂O₃, and a multi-oxide layer rich in Fe, Co, and Mn. The Ni-enriched layer maintained an FCC structure, while Cr₂O₃ exhibited a corundum structure (trigonal $\bar{R}3c$ space), and the Fe-Mn-Co layer showed a Spinel structure. The layer formation seemed to follow a selective speciation process influenced by a combination of thermodynamics and kinetics, as predicted by the PIP. Due to competition among speciating species, a single-phase Spinel-based complex oxide could have initially formed, subsequently decomposing into Fe-rich, Co-rich, and Mn-rich oxides.

To clarify the ideology behind the paper and the knowledge gap in the literature, the following has been added to the introduction:

“The structure and stability of HEAs surface oxides, however, are not well-established largely due to a wide variety of compositions and the limited understanding of the atomic scale mechanisms underlying the oxidation process. The oxide formation tendencies in a solution can thermodynamically be roughly deduced by Ellingham diagrams. However, considering many of these alloys are supersaturated-solid solutions, the concomitant oxidation and phase transformation at elevated temperatures are difficult to unveil^{2,8,9}. Application of such compositionally complex systems demands a multiscale analysis of the degradation mechanism underpinning their structural instability.”

The reviewer suggested conducting glancing incidence XRD and comparing the results with thermodynamic predictions. We appreciate that suggestion. We collaborated with National Synchrotron Light Source II, Brookhaven National Laboratories and were able to conduct grazing wide-angle X-ray scattering (GIWAXS) experiments. These experiments further confirm the presence of structurally different layers in the oxide film. Additionally, the results also reveal crystallographic details regarding the change in structure and lattice parameters when transitioning from the Ni-enriched layer to the based metal.

The latest results are added to the revised manuscript and a summary is also provided below:

Fig. 2. Ex situ Analysis of oxidized HEA (CoCrFeNiMn). (a) inverse pole figure map showing the randomly oriented recrystallized microstructure of the starting alloy condition. The inset shows the selected area diffraction pattern further confirming the FCC structure and single phase (b) A cross-section of the sample extracted using FIB after oxidation at 400°C for 120 mins at 10 mbar pressure inside the custom-made atom probe reactor chamber. The figure shows the STEM image and the elemental distribution in the oxide film and the base metal (c) STEM image further magnifying the oxide film where an arrow is used to traverse from the base alloys to the oxide film. (d) TEM EDS linear profile of elemental distribution on traversing from base metal to oxide film. (e)GIWAXS traces from $q=2.0$ - 2.7 \AA^{-1} and (f) from 2.75 - 3.25 \AA^{-1} with incidence angle varying from 0.05° (lower) to 0.5° (top). The lowest angle grazes the topmost oxide layer showing the presence of a Spinel structure and Corundum structure and as the grazing angle increases these phases disappear. Peak shift and peak splitting can be seen in (f) due to compositional changes in the matrix near the oxide metal interface.

“To conduct a comprehensive evaluation and examine the structural modifications in the oxide film, circular averages of Grazing Incidence Wide Angle X-ray Scattering (GIWAXS) data for oxidized samples are presented (see Figure S3-S4, methods, and Fig. 2 (e-f)). The lower q values reveal the presence of the Spinel structure (associated with Fe/Mn/Co oxide) and Corundum structure (corresponding to Cr_2O_3) as the outermost layer of the oxide, aligning with our TEM observations. Apart from detailing the oxidized layer's structure, noteworthy insights into the enriched layer and matrix are also provided (Fig. 2f).

“To conduct a comprehensive evaluation and examine the structural modifications in the oxide film, circular averages of Grazing Incidence Wide Angle X-ray Scattering (GIWAXS) data for oxidized samples are presented (see Figure S3-S4, methods, and Fig. 2 (e-f)). The lower q values reveal the presence of the Spinel structure (associated with Fe/Mn/Co oxide) and Corundum structure (corresponding to Cr_2O_3) as the outermost layer of the oxide, aligning with our TEM observations. Apart from detailing the oxidized layer's structure, noteworthy insights into the enriched layer and matrix are also provided (Fig. 2f).

The peak, corresponding to FCC (111) of the Ni-rich layer, shifts from $q=3.1$ to 3.06 \AA^{-1} (2θ shift from 44.7° to 44°) with an increasing incidence angle, indicating a progressive increase in lattice parameter from the top to the bottom region of the Ni-rich layer. Additionally, another FCC peak (at 3.01 \AA^{-1} or $\sim 2\theta=43.3^\circ$) starts emerging at a deeper grazing angle, corresponding to the matrix (see the unoxidized sample in the inset) This observation suggests the formation of a new FCC phase with a distinct lattice parameter between the oxide and base interface. Notably, this finding aligns with our TEM results illustrating a reoriented interface (Fig. S2).”

Furthermore, using Calphad for these predictions may not be accurate since most models in the database are tailored for high-temperature phase fields. At lower temperatures, where processes slow down, enabling the observation of various thermodynamically driven steps with spatial and temporal resolution, there is some correlation between oxide layering and thermodynamics. Yet, deviations from predictions suggest kinetic influences. To address these discrepancies, we developed a more sophisticated tool in the form of PIP.

Ellingham diagram, the standard free energies of oxide formation at 400°C rank as $\text{Al}<\text{Mn}<\text{Cr}<\text{Fe}<\text{Ni}\approx\text{Co}$.

The normalized PIP plot is displayed above. To emphasize, the advantages of PIP stem from its consideration not only of a single thermodynamic factor but of three distinct aspects:

External (E₀): This denotes the tendency to exchange electrons with the surroundings, specifically with oxygen in this case. A lower value indicates a higher likelihood of electron exchange.

Internal (CED): Similar to binding energy, this factor represents the energy required to break away from the bulk. A lower value suggests easier detachment.

Diffusion (radius): This factor indicates the mobility of the diffusing atom, with D being inversely proportional to the radius. Smaller atoms exhibit higher mobility, making them easier and faster to diffuse.

The normalized plot provides a PIP scale that extends beyond High Entropy Alloys (HEA), encompassing all elements in the periodic table. For instance, examining only the E₀ provides information akin to the Elingham diagram, where Mn's E₀ scale is at 55 and Cr is at 62. However, when combining all three factors, Cr is positioned at 2.21, and Mn at 2.24.

Additionally, the normalized plot allows for the incorporation of multiplication factors for each of these aspects, providing flexibility based on the perceived importance of each factor. Currently, they are treated as equals.

According to the PIP values:

Al > Ni > Cr > Mn > Fe > Co

1.92 > 2.15 > 2.21 > 2.24 > 2.33 > 2.47

The anomaly observed in nickel is discussed in relation to structural/phase stability, which is currently not accounted for in the PIP.

REVIEWERS' COMMENTS

Reviewer #1 (Remarks to the Author):

The authors have successfully addressed all of my questions during the revision.

Reviewer #2 (Remarks to the Author):

The authors addressed all the comments made. I believe the paper can be accepted for publication in the current view.